

# The relationship between intraseasonal tropical variability and ENSO simulated by the CMIP5

Tatiana Matveeva[1], Daria Gushchina[1]

[1]Faculty of Geography, Moscow State University, GSP-1, 119991, Leninskie Gory, Moscow, Russia

*Correspondence to*: Tatiana Matveeva (matania.777@gmail.com)

**Abstract.** This study evaluates the simulation of relationship between intraseasonal tropical variability (ITV) and El Niño Southern Oscillation (ENSO) in 23 models from the Coupled Model Intercomparison Project (CMIP) phase 5 (CMIP5) in the Intergovernmental Panel on Climate Change (IPCC) Fifth Assessment Report (AR5). As a first step, the models' skill in simulating ENSO diversity is assessed, which indicates that 16 models among 23 are able to simulate realistically the statistics of the relative size of two types of El Niño. The characteristics of the ITV are then documented revealing that only five models (CMCC-CM, CCSM4, BNU-ESM, INMCM4 and MIROC5) simulate realistically the parameters crucial for proper reproducing of ITV contribution to the El Niño, in particular the total variability, seasonal cycle and propagation along the equator of Madden-Julian oscillation (MJO) and convectively coupled equatorial Rossby waves (ER). At last step the ITV/ENSO relationship in the models are analyzed and compared to observation. It is shown that the key aspects of this interaction such as phase lag between ITV peak activity and El Niño peak and longitude localization of maximum correlation between ITV and ENSO is realistically simulated by CMCC-CM and MIROC5 for MJO and CMCC-CM and INMCM4 for equatorial Rossby waves. These models are capable to reproduce the distinct MJO and ER behavior associated to the two El Niño flavors. Aforementioned models may be used for the investigation of the sensitivity of the ITV/ENSO seasonal dependence to global warming.



# 1 Introduction

El Niño Southern Oscillation is a dominant mode of climate variability at interannual time-scale (Bjerknes 1969; Philander 1990; Neelin et al. 1998; Glantz et al. 1991; Wallace et al., 1998; Gill, 1980; Trenberth et al. 1998; McPhaden et al., 2006). It originates in the Equatorial Pacific and induces the important climate and weather anomalies in many part of the globe

through teleconnection mechanism. Therefore predicting El Niño occurrence and amplitude, both in current condition and for the next century is a key societal need (Guilyardy, 2006). The coupled ocean-atmosphere models in a wide range of complexity from "Earth system models" up to intermediate coupled model have demonstrated encouraging skill in ENSO forecast (http://iri.columbia.edu/our-expertise/climate/forecasts/enso), while simple model and observation networks were instrumental in clarifying the basic mechanisms and feedbacks at play during an El Niño (Wang and Picaut, 2004; Jin, 2006).

However the diversity of observed events as well as ENSO irregularity still poses a serious barrier for further improvement of El Niño forecast. The latter is largely attributed to the stochastic atmosphere forcing (McWilliams and Gent, 1978; Lau, 1985; Penland and Sardeshmukh, 1995; Penland, 1996; Blanke et al., 1997; Kleeman and Moore, 1997; Eckert and Latif, 1997; Moore and Kleeman, 1999; Thompson and Battisti, 2001; Dijkstra and Burgers, 2002; Larkin and Harrison, 2002; Kessler, 2002)

Recent investigation evidenced the pivotal role of intraseasonal tropical variability (ITV) in triggering of El Niño. The dominant intraseasonal mode in tropics – the Madden-Julian Oscillation (MJO) – was shown to be tightly related to ENSO through its relationship to episodes of westerlies that can trigger downwelling intraseasonal Kelvin waves, a precursor to El Niño onset (McPhaden et al., 2006; Hendon et al., 2008; Gushchina and Dewitte, 2011, 2012; Puy et al., 2015). However MJO is not the only important component of the ITV involved in the ENSO generation mechanism. Puy et al. (2015)

highlighted the role of equatorial Rossby (ER) wave in the generation of Westerly wind events (WWE) characterized as episodes of anomalous, short-lived, but strong westerlies developing over the western Pacific warm pool (e.g. Luther et al. 1983). These WWEs promote the onset and development of El Niño events (Fedorov, 2002; Lengaigne et al., 2003; Boulanger et al., 2004) and contribute to the irregularity of ENSO. Gushchina and Dewitte (2012) suggested that the activity of equatorial Rossby waves is associated to the amplitude of oceanic Kelvin wave. While the anomalous westerlies related to

the convective phase of MJO induce the oceanic Kelvin wave in the Western Pacific in March-April preceding El Niño peak,



the intensification of convectively coupled equatorial Rossby waves in June-July in the central Pacific acts to compensate for the Kelvin wave dissipation along its way through the eastern Pacific.

The diversity of El Niño events may be considered in terms of obvious decadal modulation of ENSO cycle (Trenberth and Shea, 1987; Cobb et al., 2003; An, 2004; An and Jin, 2004). Recently it was emphasized that El Niño appears in two flavors:

central Pacific (CP) El Niño with SST anomalies localized in the vicinity of the date line and Eastern Pacific (EP) El Niño characterized by SST anomalies confined to the tropical eastern Pacific (Kug et al., 2009; Capotondi et al., 2015). The induced atmosphere response differs drastically between two types of El Niño (Ashok et al., 2007; Weng et al., 2008; Frauen et al., 2014; Zheleznova and Gushchina, 2015, 2016). It was shown that the statistics of the diversity of ENSO is modulated at decadal scale with a tendency for more occurrence of CP events since 2000 (Yeh et al., 2009; Lee and McPhaden, 2010;

Takahashi et al., 2011; Cai et al., 2015). This raises question about the influence of mean state change, in particular the global warming, on the regimes of El Niño (Capotondi et al., 2015). Yeh et al. (2009) highlighted an increase (decrease) of the occurrence of CP (EP) El Niños during the recent decades, suggesting this may result from global warming. The change of mean state was shown to impacts also the predictability of ENSO. McPhaden (2012) noted that major source of ENSO predictability – the warm water volume (WWV) over the equatorial Pacific – may have loose some of its predictive values

since the 2000. Gushchina and Dewitte (2017) have documented that ENSO/ITV relationship does not only have a marked seasonal dependence, it is also sensitive to state of the tropical Pacific, which has implication for ENSO seasonal forecasts. This raises concerns on how the ITV/ENSO relationship may change in the future climate. This study implies the use of climate model data. However recent investigation demonstrated that the diversity of El Niño as well as main characteristics of ITV components are poorly reproduced by several climate models.

Yu and Kim (2010) analyzed the reproducing of two types of ENSO in the Coupled Model Intercomparison Project (CMIP) phase 3 (CMIP3).They documented that most CMIP3 models (13 out of 19) can produce realistically strong CP ENSOs, but only a few of them (9 out of 19) can produce realistically strong EP ENSOs. Only six models realistically simulate the intensity ratio between EP and CP ENSOs. CMIP phase 5 (CMIP5) generation models have demonstrated significant improvements in simulation of ENSO types (Kim and You, 2012). Firstly, the simulated spatial patterns of both types of

ENSO are closer to the observed one. Secondly, the inter-model differences in the CP and EP events intensity is reduced in

CMIP5 as compare to CMIP3 models. The decrease in the inter-model discrepancies is more pronounced for EP event. However 50% of the CMIP5 models still cannot simulate realistically strong CP and especially EP El Niños.

Hung et al. (2013) evaluated the simulation of the Madden–Julian oscillation and convectively coupled equatorial waves (CCEWs) in 20 models from CMIP5 and compares the results with the simulation of CMIP3 models (Lin et al., 2006). It was

demonstrated that CMIP5 models exhibit an overall improvement in the simulation of tropical intraseasonal variability, especially the MJO and several CCEWs as compare to CMIP3 models. The CMIP5 models produce larger total intraseasonal variance of precipitation than the CMIP3 models, including larger variances of MJO, Kelvin, equatorial Rossby (ER), and eastward inertio-gravity (EIG) waves. About one-third of the CMIP5 models generate the spectral peak of MJO precipitation between 30 and 70 days; however, the model MJO period tends to be longer than observations and only one of the 20 models

is able to simulate a realistic eastward propagation of the MJO. In current research the attention is drawn to the ITV characteristics crucial for interaction with ENSO, in particular intensity, seasonal cycle, longitude position and propagation speed.

While the ITV and ENSO simulation in CMIP5 has been documented, there is no documentation of how the ITV contribution to the El Niño cycle is captured by the models. Therefore the purpose of this study is to select the models the

most skilful in simulation of MJO and Rossby waves contribution to the El Niño generation and therefore promising for investigation of the sensitivity of the ITV/ENSO seasonal dependence to global warming.

The models and validation datasets as well as the diagnostic methods used in this study are described in section 2. The simulation of two types of El Niño, ITV components and ITV/ENSO relationship in CMIP5 models are analyzed in section 3. A summary and discussion are given in section 4.

**2 Models and datasets**

**2.1 Data**

The data of 23 models from the Coupled Model Intercomparison Project (CMIP) phase 5 (CMIP5) in the Intergovernmental Panel on Climate Change (IPCC) Fifth Assessment Report (AR5) is used (Table 1).



To analyze the SST distribution associated to the ENSO events the monthly SST anomalies over 250-years of simulation are calculated. The Hadley Centre Global Sea Ice and Sea Surface Temperature (HadISST, Rayner et al. (2003)) archive is used to derive the observed monthly SST anomalies.

The daily data of zonal wind at 850 hPa for 20-years period from model simulations and NCEP/NCAR Reanalysis is used to

isolate the MJO and CCEWs.

For model analysis the data of PI-Control experiment was used with preindustrial concentration of greenhouse gases. The Historical experiment better reproduces the modern climate condition and is commonly used for model validation. However this study aims to select the models, the most successful in simulation ENSO/ITV relationship, for further investigation of its modification under climate warming. As the model diversity in assessment of ENSO modification in future climate is rather

large (Taschetto et al., 2014; Chen et al., 2017; Xu et al., 2017) it is preferably to compare the most contrasting scenarios, i.e PI-Control with any external forcing and RCP8,5 with the strongest radiation forcing. Therefore we need to evaluate the CMIP5 model skill in reproducing El Niño diversity as well as ENSO/ITV relationship in experiment PI-Control.

## 2.2 Methods

To document the ITV patterns we use the technique proposed by Wheeler and Kiladis (1999). This method is identical to

those used in the previous studies evaluating the simulation of MJO and CCEW in CMIP3 (Lin et al. ,2006) and CMIP5 (Hung et al., 2013) models. It is based on the decomposition of the symmetric and antisymmetric relative to the equator components of the zonal wind at 850 hPa (U850) in the frequency-wavenumber space. Inversed Fourier transform is then used to recompose the signal in the desired frequency and wavenumber bands. The frequency and wavenumber intervals were derived from the normalized space-time spectrum for U850 and are centered on the spectral maximum of U850 (cf.

Gushchina and Dewitte (2011)). These are for MJO – zonal wavenumber 1-3, period 30-96 days, for equatorial Rossby waves - zonal wavenumber -1…-8, period 10-50 days, for Kelvin waves zonal wavenumber 1-9, period 3-30 days (with negative (positive) zonal wavenumber corresponding to the westward (eastward) propagating waves). The amplitude of equatorial waves and MJO was calculated by taking the root mean square (rms) of the MJO, Equatorial Rossby and Kelvin waves filtered U850, with the rms computed in a running window dependent on the wave's type (90 , 48 and 30 days for

MJO, Equatorial Rossby and Kelvin waves respectively). Then the running rms was monthly averaged. The amplitude time series were first equatorially averaged (5°S-5°N) at each point of longitude. To obtain the indices of ITV activity, the amplitude time series were averaged over the regions where the maximum of ITV/ENSO relationship is observed (see Table 2). These indices are further referred as MJO and Rossby wave indices - $WPacMJO_{u850}$ and $CPacER_{u850}$.

For correlation analysis between ITV activity and ENSO types the so-called E and C indices based on the linear combination of the first two EOFs of the SST anomalies in the tropical Pacific were constructed (Takahashi et al., 2011). Whereas the E index accounts for the extreme El Niño events (EP type), the C index grasps the variability associated to the CP El Niño and La Niña events. These indices, independent by construction (i.e. their correlation is zero), can be successfully used for correlation analyses (see Takahashi et al. (2011) for more details).

# 3 Results

## 3.1 The two flavors of El Nino

As a first step, the models' skill in simulating ENSO diversity is assessed. Takahashi et al. (2011) demonstrated that the patterns of the first two EOF of SST anomalies in the tropical Pacific capture the structure of SST anomalies associated to the two types of El Niño, with the EOF1 maximum localized in the eastern Pacific (typical of EP event) and EOF2 maximum observed in the central Pacific (typical of CP event). The two first EOF modes obtained from 23 CMIP5 models are presented on Fig. 1-2.

Almost all the models (except for CSIRO-Mk3) reproduce the SST anomalies pattern associated to the EP El Niño (Fig.1) with the maximum localized in the eastern equatorial Pacific. The main deficiency of the models is the westward shift of anomaly maximum as compare to observation. Almost half of models are unable to simulate adequately the tripole structure of CP El Niño with maximum in the central Pacific bordered by two centers of opposite sign (Fig. 2). Noteworthy that most models tend to overestimate the variability associated to the first EOF as compare to observation. In opposite the EOF2 contribution to the total variability is lower in the model than in HadISST data (except for ACCESS1-3, CSIRO-Mk3, FIO-



ESM и INMCM4). Therefore we may suggest that models tend to underestimate the CP El Niño contribution to the SST variability in the tropical Pacific.

To demonstrate the models skill in reproducing ENSO period the spectrum of C and E El Niño indices is presented (Fig.1-2). Most models simulate several distinct spectral peaks in the interval 3-9 years. The E index peaks are shifted to the shorter

periods (3-6 years) while CP El Niño is characterized by longer oscillation (5-8 years), which is in a good agreement with observation (4-5 and 5-6 years for EP and CP events respectively). The models tend to overestimate the spectral density which is partially due to the longer investigated period for the model as compare to HadISST (250 and 135 years respectively).

Based on the EOF and spectral analysis the 16 models capable to reproduce the spatial structure and temporal variability of

SST associated to the two distinct type of El Niño were chosen for further analysis: BNU-ESM, CanESM2, CCSM4, CESM1-CAM5, CMCC-CM, CNRM-CM5, EC-EARTH, FIO-ESM, GFDL-CM3, GFDL-ESM2M, GISS-E2-H, INMCM4, IPSL-CM5A-MR, MIROC 5, MPI-ESM-LR, MRI-CGCM3.

### 3.2 Intraseasonal tropical variability

The characteristics of the ITV are then documented with the focus on the key aspects for relationship between ENSO and

ITV, namely the intensity, spatial structure, propagation characteristics, frequency and seasonal cycle of ITV component. The aforementioned characteristics primarily determine the efficiency of ITV contribution to the El Niño development.

The earlier studies have evidenced inaccurate simulation of MJO and CCEW in CMIP models (Guo et al., 2015; Jiang et al., 2015; Klingman et al., 2015; Xavier et al., 2015), with CMIP5 demonstrated more skill (Hung et al., 2013) than CMIP3 generation (Lin et al., 2006). We tested the models reliable in reproducing two type of El Niño to simulate adequately the

main ITV components: MJO, equatorial Rossby and Kelvin waves. We had to exclude from the analysis the models which do not present in open access the daily data of U850 required for ITV filtering. The 20-years period was used for the analysis as most of the models have not longer period of daily data. The following models were analyzed (with the period of model years indicated in the parenthesis): BNU-ESM (1979-1998), CanESM2 (2441-2460), CCSM4 (1089-1108), CMCC-CM (1850-1869), CNRM-CM5 (2385-2404), GFDL-CM3 (0001-0020), GFDL-ESM2M (0001-0020), INMCM4 (2090-2109),



IPSL-CM5A-MR (1800-1819), MIROC 5 (2000-2019), MPI-ESM-LR (2016-2035), MRI-CGCM3 (2086-2105). The model outputs are compared to the NCEP/NCAR Reanalysis data for the period 1980-1999.

To isolate the ITV component the raw wavenumber-frequency spectra were plotted following the method of Wheeler and Killadis (see Section 2). Figures 3 present the space-time spectra normalized above the background spectra for symmetric

component of U850 wind from the Reanalysis over 1980-1999 (Fig.3 upper panel) and as simulated by CMIP5 models over a 20-year period. The resulting contours can be thought of as levels of significance, with peaks in the individual spectra that are 20% above the background shown as shaded. Superimposed upon these plots are the dispersion curves for odd meridional mode number of equatorial waves for various equivalent depths (h=8, 12, 25, 50, 90 and 200 m). The MJO appears as a prominent signal, especially in the symmetric spectra (the spectra for antisymmetric component is not shown).

Most of the models simulate the MJO signal, but 8 among 12 overestimate the westward propagating signal against eastward one. Signals of the Kelvin waves are obviously identified in the symmetric spectra for observation. The main model deficiency in representation of Kelvin wave is the unrealistic spectral maximum around eastward wave numbers 6-8. The simulated signal appears in both the symmetric and antisymmetric spectra and was shown to be associated to the mid latitude waves instead of Kelvin waves (Gushchina and Dewitte, 2011). Several models simulate low spectral energy in Kelvin wave

interval. The ER signal is lower than MJO and Kelvin, it does not exceed by 20% the background spectra, but may be identified in the symmetric spectra, both for observation and models.

The total variance of MJO and ER is of particular importance for correct simulation of ITV forcing of the equatorial ocean. To estimate the total variance associated to the MJO, ER and Kelvin waves the rms over 20 year period averaged between 15°N and 15°S were plotted as function of longitude for the models and compare to the results obtained from NCEP/NCAR

Reanalysis (Fig.4). The localization of MJO maximum in the eastern Indian ocean and western Pacific is captured by CCSM-4, CMCC-CM, MIROC5 and BNU-ESM models. Other models do not exhibit a significant peak in this region which may be critical for proper simulation of ITV forcing of oceanic Kelvin wave. Only 5 models out of 12 demonstrate the reasonable magnitude and longitudional distribution of ER variance, while the maximum in the central Pacific is correctly reproduced by MPI-ESM-LR model only. The modelled Kelvin wave variance is more realistic, however half out of 12

models tend to underestimate the total Kelvin wave amplitude.





In (Hendon et al., 2007) the important seasonal dependence of the lagged association of the MJO with ENSO was documented. The seasonally varying relationship between MJO activity and the ENSO cycle is due to the marked seasonal cycle of the MJO activity and phaselocking of ENSO onset to the seasonal cycle. As was suggested the anomalous westerlies associated to the MJO induce the oceanic Kelvin wave in the Western Pacific in March-April preceding El Niño. The intensification of equatorial Rossby waves (ER) in June-July in the central Pacific serves the maintenance for Kelvin wave dissipating along its way through the Pacific (Gushchina and Dewitte, 2011, 2012). The latter being responsible for anomaly propagation along the equatorial Pacific and resulting in El Niño conditions. As the oceanic Kelvin wave is strongly confined to the equatorial belt, the MJO and Rossby waves may induce the oceanic wave only in the near equatorial region. The maximum of Rossby wave is located on the equator but may improve the seasonal variation of intensity. The maximum of MJO is attributed to the summer hemisphere and displays the pronounced seasonal zonal migration. In boreal spring the MJO has maximum intensity on the equator and may acts efficiently as an ENSO trigger. Therefore the model's capability to produce the MJO crossequatorial migration as well as MJO and ER seasonal cycle is crucial for proper simulation of El Niño onset. The ITV seasonal variability were estimated over tropical Pacific (120°E-90°W) for MJO along 3 latitude belts: 10°N-15°N, 5°N-5°S и 10°S-15°S (Fig. 5) and along the equator for Rossby wave (fig.6). In NCEP/NCAR reanalysis MJO improves larger variability in summer hemisphere with higher amplitude in Southern than in Northern hemisphere. In northern tropical Pacific (10°N-15°N) the MJO activity peaks from June to September (Fig. 5a,b). In near equatorial area no marked seasonal peaks are observed, with slight intensification from November to April and relaxation from May to October (Fig. 5c,d). The maximum variability of MJO is attributed to the Southern Hemisphere (Fig. 5e,f), with rms maximum occurs from November to March. The seasonal shift of MJO maximum drastically differs between the models. The comparison of model output with observation evidences the realistic MJO seasonal cycle reproduced by CMCC-CM, CCSM4 and MIROC5 models. BNU-ESM and INMCM4 models demonstrate the correct timing of seasonal maximum with underestimated MJO amplitude. The seasonal cycle of ER intensity is captured by CMCC-CM, CCSM4, BNU-ESM, INMCM4 and MIROC5. Noteworthy the variability of equatorial Rossby wave in these models is close to the observed one (Fig.6).

The characteristic of MJO and ER propagation along the equator are then documented for models and observation. The models that demonstrated proper simulation of ITV variability and seasonal cycle are analyzed (Table 3).

The propagation velocity and spatial patterns of MJO are adequately simulated by the models (Fig. 7, Table 3), with however faster propagation in CCSM4 (Fig. 7c) and shorter wave length and lower speed in BNU-ESM (Fig. 7b). The equatorial Rossby waves have larger longitudional extent in the models (Fig. 8). The propagation velocity is comparable to Reanalysis in BNU-ESM (Fig. 8b), CMCC-CM (Fig. 8d) and INM-CM4 (Fig. 8e), and is too low in CCSM4 (Fig. 8c) and MIROC5

(Fig. 8f). In INMCM4 and MIROC 5 the intensity of Rossby waves is overestimated. The spatial structure and propagation velocity of Kelvin waves are comparable to the observation for most models, except for INMCM4 with low amplitude of Kelvin waves (not shown).

### 3.3 ITV/ENSO relationship

While several studies suggest the predictive value of the ITV for ENSO (McPhaden et al., 2006; Hendon et al., 2007;

Gushchina and Dewitte, 2011; Puy et al., 2015), Gushchina and Dewitte (2012) indicated a distinct ENSO/ITV seasonal relationships according to the El Niño type. Namely, during EP El Niño the MJO and equatorial Rossby waves act as a trigger of the event while during Modoki El Niño they contribute also to its persistence once it has appeared. Therefore it is of primary importance to analyze the CMIP5 models skill in simulation different MJO and ER behavior associated to the two flavors of El Niño.

On the Figures 9, 11 the lag-correlation between C and E indices (see section 2) in January and the equatorially averaged wave amplitude for U850 is presented as a function of longitude for NCEP/NCAR Reanalysis and 5 models (CMCC-CM, CCSM4, BNU-ESM, INMCM4 and MIROC5), that were shown to simulate successfully the ENSO diversity and ITV characteristics. The reference month for the ENSO indices is January (0) which corresponds to the approximate time of the El Niño peak. The lag-correlation of the C and E indices with respect to the MJO and ER indices (see Section 2) is presented

as a function of calendar month (start month) for CMCC-CM, INMCM4 and MIROC5 models, which are the most skillful in simulating the relationships between equatorial ITV and SST at ENSO peak (Fig. 10, 12).BNU_ESM is rejected as it reproduces unrealistic relationship between EP event and ITV, while CCSM4 does not simulate CP/ITV interaction. Correlation higher than 0.42 is significant at 90% significance level assuming Gaussian statistics with 20 independent





samples. The model results are compared to NCEP/NCAR Reanalysis. The analyzed period for Reanlaysis is 1979-1998 for EP El Niño and 2000-2015 for CP El Niño, considering an increased occurrence of CP events since 2000.

### 3.3.1 MJO

Based on Reanalysis data the MJO activity may be considered as a precursor of El Niño events: MJO intensification leads

the E (C) January index by 9-4 (8-2) months (Fig. 9a,g). The MJO forcing in the western Pacific during Eastern Pacific El Niño (intensification in spring-summer prior to Jan0) is captured by CMCC-CM (Fig. 9d), INMCM4 (Fig. 9e) and MIROC5 (Fig. 9f) models. However the MJO intensification in CMCC-CM (Fig. 9d) occurs later in seasonal cycle (in September-November as compare to February-May in Reanalysis) and earlier in seasonal cycle in MIROC5 (Fig.9f). In INMCM4 the maximum MJO intensification is shifted to the Indian ocean (Fig. 9e). For CP El Niño the MJO precursor signal similar to

the observed one appears in BNU-ESM (Fig. 9g) and CMCC-CM (Fig. 9j) models. In CCSM4 and INMCM4 (Fig. 9i,k) the signal appears earlier and in MIROC5 (Fig. 9l) later in seasonal cycle. Noteworthy almost all model able to simulate the intensification of MJO during Central Pacific El Niño decaying phase, the latter being responsible for the persistence of SST anomalies during CP event. However the correlation is lower than in observation.

The analysis of MJO/ENSO relationships as a function of start month (Fig. 10) suggests the mostly reliable models are

CMCC-CM (Fig.10b,f) and MIROC5 (Fig.10d,h). The MJO activity in surface winds in the western Pacific in March–July highly correlates (correlation greater 0.6) with SST anomalies 4-12 (3–9) months later during EP (CP) El Niño in NCEP/NCAR data (Fig. 10a,e). The significant positive correlation persists up to negative time-lags (MJO lags SST) during CP event mirroring the strong MJO after the SST peak. The MJO intensification during development phase of EP event (positive time lags) is simulated by MIROC5 (Fig.10d), however it occurs earlier in the year (January-May) as compare to

observation. MIROC5 reproduces unrealistic MJO activity after SST maximum. The precursor signal for CP event is too weak in MIROC5 (Fig.10h). CMCC-CM captures the overall structure of correlation patterns for both type of El Niño events (Fig.10b,f), but the precursor signal appears too late (since August).



### 3.3.2 Equatorial Rossby waves

Enhanced ER activity is also found prior to both El Niño types with contribution comparable to MJO (Gushchina and Dewitte, 2011; Puy et al., 2015). Equatorial Rossby waves intensify in the central equatorial Pacific in June-November preceding EP El Niño culmination (Jan0) (Fig. 11a). This pattern is simulated by CCSM4 (Fig.11c), CMCC-CM (Fig.11d),

INMCM4 (Fig.11e) and MIROC5 (Fig.11f) models, however in MIROC5 the intensification appears earlier in seasonal cycle and in INMCM4 and CCSM4 the magnitude of correlation is too weak. During CP El Niño Rossby wave activity positively correlate with SST in January since March up to December and stays important after the culmination phase (up to August of the next year). CCSM4 (Fig.11i) and INMCM4 (Fig.11k) models are not successful in simulating the ER/CP relationships. In MIROC5 (Fig.11l) the intensification occurs later in seasonal cycle, while in BNU-ESM (Fig.11h) and

CMCC-CM (Fig. 11j) the timing of precursor signal is closer to observed one but the correlation is lower than in observation.

The correlation of E index with PacERu850 demonstrates the ER intensification in February-April and July-September with successive raise of SST in the eastern Pacific 12-6 and 8-2 months later respectively and abrupt decay after the culmination for EP event (Fig. 12a). During CP El Niño the Rossby wave activity remains after the mature phase (Fig. 12e). CMCC-CM

(Fig.12b) is very close to observation for EP event. During CP El Niño the correlation is lower in the model as compare to observation and the first correlation maximum associated to the ER intensification in February-April prior to El Niño is not captured by the model. INMCM4 demonstrates a good accordance with Reanalysis in reproducing ER intensification at EP El Niño development phase (Fig. 12c), but the strong ER activity remains after the peak of the event, while it weakens in observation. For CP El Niño INMCM4 is less successful (Fig. 12g): the precursor signal is presented but has lower

amplitude and appears at smaller time lag as compare to observation which diminishes its predictive value in the model. In MIROC5 the strong persistent signal appears in May-August during EP event (Fig. 12d) and in September-November during CP event (Fig. 12h) with no counterpart in observation. Overall the less accordance between models and observation during CP El Niño may be due to the fact that for observation the period 2000-2015 is used with increased occurrence of CP events. Meanwhile in the model the same 20-years period was used for E and C index with similar occurrence of EP and CP events.

### 3.3.3 Equatorial Kelvin wave

The intensification of atmosphere Kelvin equatorial waves occurs during ENSO culmination phase and represents the response of tropical atmosphere to the anomalous heat source related to the latent heat release in system of deep convection (Gill, 1980).Therefore Kelvin waves do not exhibit the predictive value relative to ENSO. The Kelvin/ENSO relationships

are fairly reproduced by CMIP5 models (not shown).

### 4. Summary

In this study we evaluate the simulation of relationship between intraseasonal tropical variability (ITV) and El Niño Southern Oscillation (ENSO) in 23 models from the Coupled Model Intercomparison Project (CMIP) phase 5 (CMIP5) in

the Intergovernmental Panel on Climate Change (IPCC) Fifth Assessment Report (AR5).

The models' skill in simulating ENSO diversity is assessed. 16 models are demonstrated to separate Eastern Pacific and Central Pacific El Niño with common deficiency of the models to shift the maximum of SST anomalies to the west. Most models tend to overestimate (underestimate) the variability explained by the EOF1 (EOF2), the first (latter) being associated to SST pattern typical for EP (CP) event. The period of El Niño is realistic in most models with shorter periods (3-6 years) of

EP El Niño and longer oscillation (5-8 years) of CP El Niño.

The characteristics of the ITV in the models are then documented. We focused on the simulation of total variability, seasonal cycle and propagation speed of Madden-Julian oscillation (MJO) and convectively coupled equatorial Rossby waves (ER) that determine the amplitude and phaselocking of ITV/ENSO relationship. CMCC-CM, CCSM4, BNU-ESM and MIROC5 simulate the MJO variability comparable to observation with the maximum in the eastern Indian ocean and Western Pacific.

Only 5 models out of 12 demonstrate the reasonable magnitude and longitudional distribution of ER variance, while the maximum in the central Pacific is correctly reproduced by MPI-ESM-LR model only. The realistic MJO seasonal cycle with intensification on the equator in boreal spring is reproduced by CMCC-CM, CCSM4 and MIROC5 models. The correct timing of seasonal maximum with underestimated MJO amplitude demonstrate BNU-ESM and INMCM4 models. The

seasonal cycle of ER intensity is captured by CMCC-CM, CCSM4, BNU-ESM, INMCM4 and MIROC5. The propagation of the MJO and ER patterns along the equator is realistic in CMCC-CM, CCSM4, BNU-ESM, INMCM4 and MIROC5 models. The ITV/ENSO relationship are analyzed on the ground of lag correlation between indices of EP/CP El Niño and MJO/ER activity. It is shown that the key aspects of this interaction such as phase lag between ITV peak activity and El Niño peak

5    and longitude localization of maximum correlation between ITV and ENSO are realistically simulated by CMCC-CM and MIROC5 for MJO and CMCC-CM and INMCM4 for equatorial Rossby waves. Noteworthy these models captures distinct MJO and ER behavior associated to the Eastern Pacific and Central Pacific El Niño. The deficiency of INM-CM4 in simulation of MJO/ENSO relationship is due to the inaccurate simulation of seasonal cycle and underestimated amplitude of MJO. Meanwhile MIROC5 demonstrates realistic features in simulation both MJO and ER characteristics, therefore the

10   reason of its inability to reproduce the ER precursor signal is questionable and implies further investigation.

In spite of several discrepancies between the model and observation we may conclude that El Niño precursor signal induced by MJO and ER is presented in CMCC-CM, MIROC5 and INMCM4. This may involve further investigation of ITV predictive value relative to ENSO in future climate and its sensitivity to global warming using these models.

**Code availability**

15   The code on Fortran and Matlab is available on request from author (Tatiana Matveeva, matania.777@gmail.com).

**Acknowledgement**

This study is supported by grants of Russian Foundation of Basic Research No.15-05-06693 and No.16-35-00394\16. The study is carried out in frame of  scientific program of Faculty of Geography of Moscow State University No.AAAA-A16-116032810086-4.



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



Table 1. Description of the 23 CMIP5 coupled models analyzed in this study.

| Model | Modeling Group (or Center) | Atmospheric grid | |
|---|---|---|---|
| | | latitude | longitude |
| ACCESS1-3 | Commonwealth Scientific and Industrial Research Organisation/Bureau of Meteorology, Australia | 1.25° | 1.875° |
| BNU-ESM | Beijing Normal University, China | 2.7906° | 2.8125° |
| CanESM2 | Canadian Centre for Climate Modelling and Analysis, Canada | 2.8125° | 2.8125° |
| CCSM4 | National Center for Atmospheric Research, USA | 0.9424° | 1.25° |
| CESM1-CAM5 | National Science Foundation, Department of Energy, National Center for Atmospheric Research, USA | 0.9424° | 1.25° |
| CMCC-CM | Centro Euro-Mediterraneo per I Cambiamenti Climatici, Italy | 0.7484° | 0.75° |
| CNRM-CM5 | Centre National de Recherches Météorologiques, Centre Européen de Recherche et de Formation Avancée en Calcul Scientifique, France | 1.4008° | 1.40625° |
| CSIRO-Mk3 | Commonwealth Scientific and Industrial Research Organization/Queensland Climate Change Centre of Excellence, Australia | 1.8653° | 1.875° |
| EC-EARTH | EC-EARTH consortium (ECMWF consortium) | 1.125° | 1.125° |
| FIO-ESM | The First Institute of Oceanography, SOA, China | 2.8125° | 2.8125° |
| GFDL-CM3 | Geophysical Fluid Dynamics Laboratory, USA | 2° | 2.5° |
| GFDL-ESM2M | Geophysical Fluid Dynamics Laboratory, USA | 2° | 2.5° |
| GISS-E2-H | NASA/GISS (Goddard Institute for Space Studies), USA | 2° | 2.5° |



| | | | |
|---|---|---|---|
| GISS-E2-R | NASA/GISS (Goddard Institute for Space Studies), USA | 2° | 2.5° |
| HadGEM2-CC | Met Office Hadley Centre, UK | 1.25° | 1.875° |
| HadGEM2-ES | Met Office Hadley Centre, UK | 1.25° | 1.875° |
| INM-CM4 | Russian Academy of Sciences, Institute of Numerical Mathematics, Russian Federation | 1.5° | 2° |
| IPSL-CM5A-MR | Institut Pierre Simon Laplace, France | 1.2676° | 2.5° |
| MIROC5 | Atmosphere and Ocean Research Institute, National Institute for Environmental Studies and Japan Agency for Marine-Earth Science and Technology, Japan | 1.4008° | 1.40625° |
| MPI-ESM-LR | Max Planck Institute for Meteorology, Germany | 1.8653° | 1.875° |
| MPI-ESM-P | Max Planck Institute for Meteorology, Germany | 1.8653° | 1.875° |
| MRI-CGCM3 | Meteorological Research Institute, Japan | 1.12148° | 1.125° |
| NorESM1-M | Bjerknes Centre for Climate Research, Norwegian Meteorological Institute, Norway | 1.8947 | 2.5° |





Table 2. Wave indices area.

|  | WPacMJO$_{u850}$ | CPacER$_{u850}$ |
| --- | --- | --- |
| NCEP-NCAR | 5°S – 5°N, 120°E – 180°E | 5°S – 5°N, 140°E – 160°W |
| CMCC-CM | 5°S – 5°N, 160°E – 130°W | 5°S – 5°N, 160°E – 150°W |
| INM-CM4 | 5°S – 5°N, 170°E – 140°W | 5°S – 5°N, 180° – 130°W |
| MIROC5 | 5°S – 5°N, 170°E – 130°W | 5°S – 5°N, 150°E – 150°W |





Table 3. The main characteristics of MJO and ER.

| Model | MJO | | Rossby waves |
|---|---|---|---|
| | Velocity, km/day | Extension, *1000 km | Velocity, km/day |
| NCEP/NCAR Reanalysis | 300-430 | 12-20 | 420-550 |
| BNU-ESM | 200-300 | 9-12 | 475-560 |
| CCSM4 | 350 -500 | 15-22 | 340-400 |
| CMCC-CM | 350-500 | 16 -20 | 420-500 |
| INM-CM4 | 350-450 | 15-18 | 420-500 |
| MIROC5 | 300-450 | 15-19 | 300-350 |





**Figure 1: The first EOF mode of SST anomalies calculated over 20°S - 20°N, 120°E - 90°W (upper panels) and power spectral density of E-index (bottom panels) for observations (HadISST) and 23 CMIP5 models. The percentage of explained variability is indicated in parenthesis**





Figure 2: The same as Fig.1 but for the second EOF and C-index.





**Figure 3: Space–time spectrum of the 15°N–15°S symmetric component of U850 divided by the background spectrum from NCEP/NCAR Reanalysis and 12 CMIP5 models.**





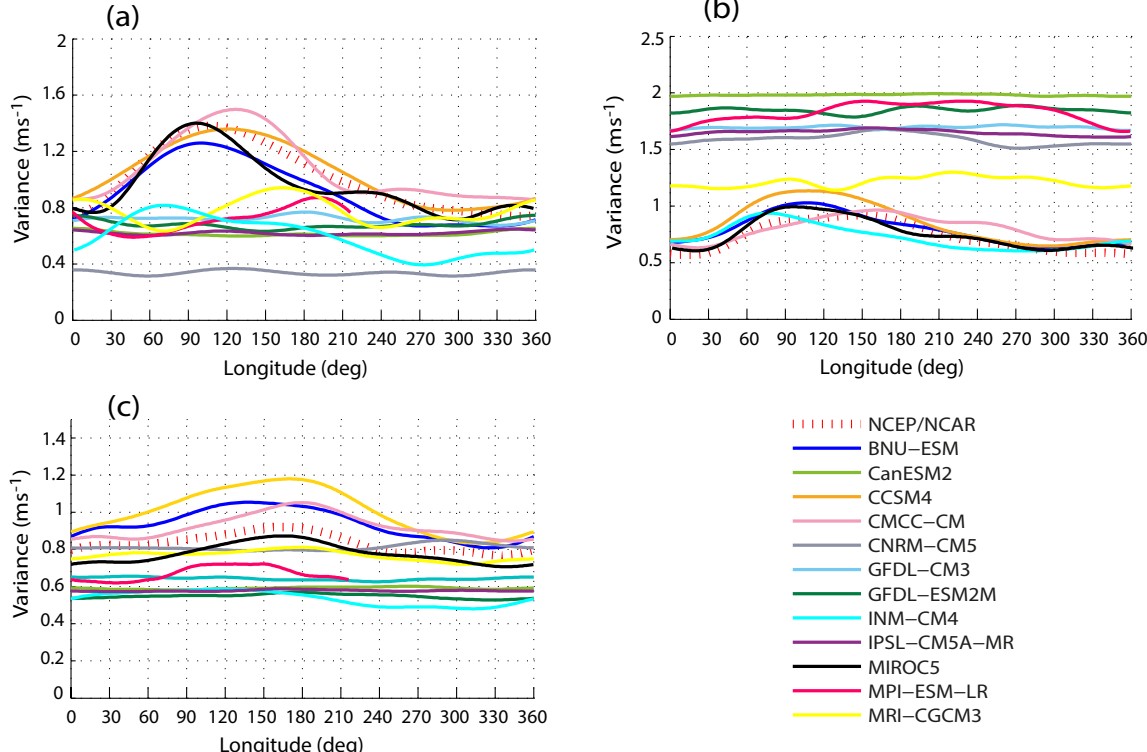

**Figure 4: Variances (rms) of MJO (a), Rossby (b) and Kelvin (c) waves along the equator averaged between 15°N and 15°S from NCEP/NCAR Reanalysis and 12 CMIP5 models**





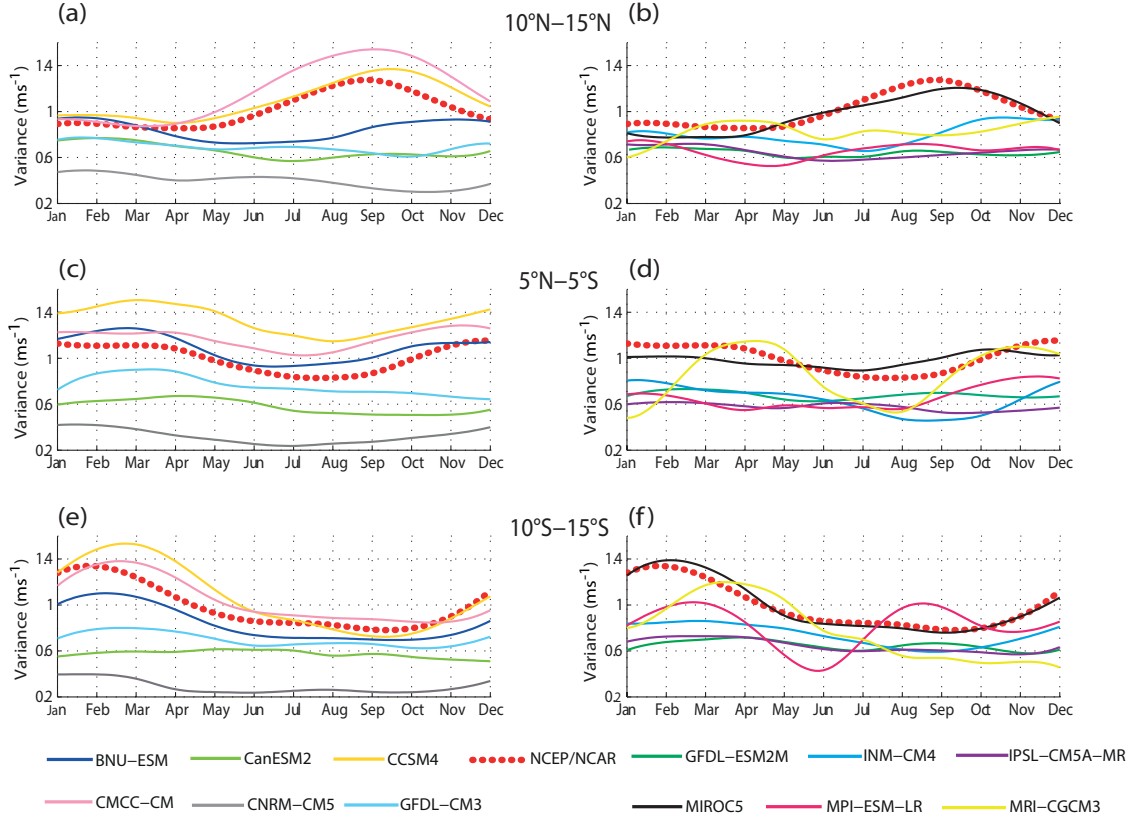

**Figure 5: Seasonal variances (rms) of MJO averaged over: 10°N-15°N (a,b), 5°N-5°S (c,d) and 10°S-15°S (e,f) from NCEP/NCAR Reanalysis and 12 CMIP5 models.**

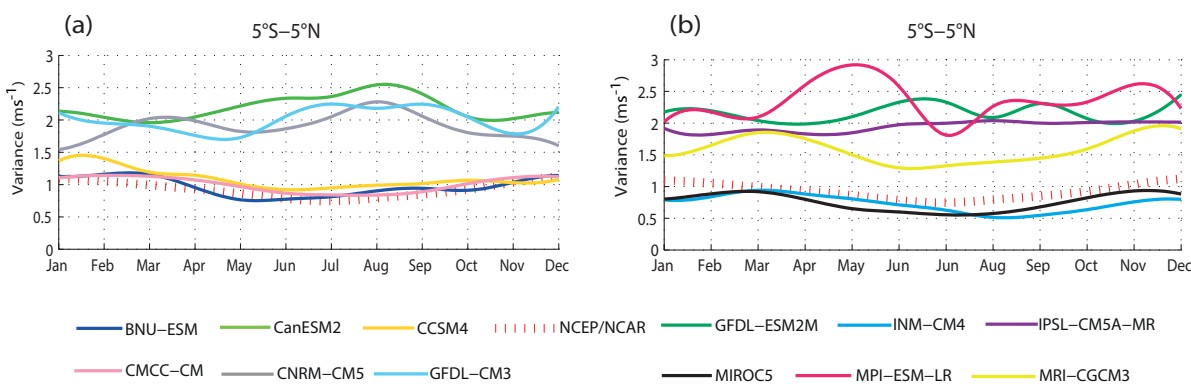

**Figure 6: Seasonal variances (rms) of Rossby waves averaged over 5°S-5°N from NCEP/NCAR Reanalysis and 12 CMIP5 models.**



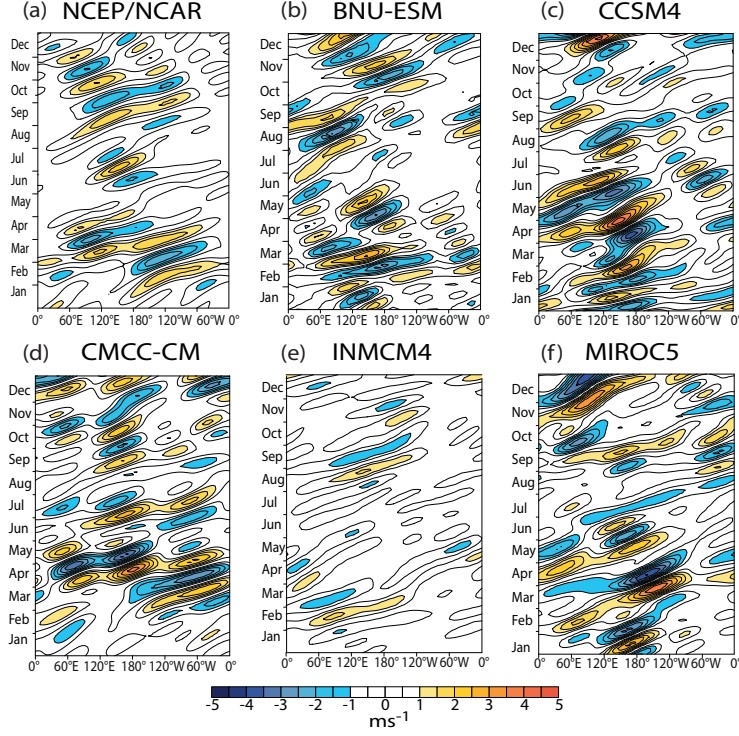

**Figure 7: Time-longitude plots of equatorial averaged (5°N–5°S) daily-mean anomalies of MJO filtered U850 from NCEP/NCAR Reanalysis and 5 CMIP5 models. Contour interval is 0.5 m/s. Negative values ≤–1 m/s are blue shaded, positive values ≥1 m/s are orange shaded.**

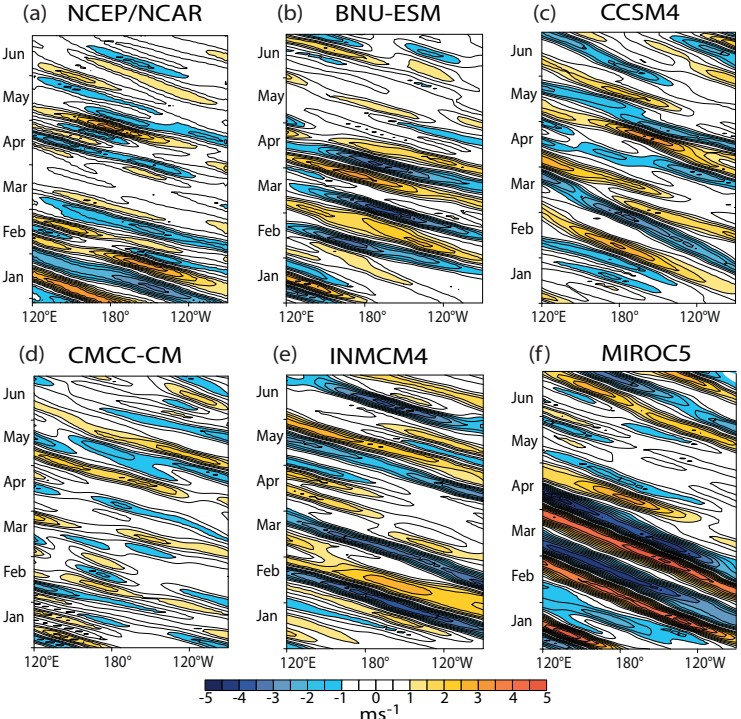

**Figure 8: As in Figure 7, but for Rossby waves.**



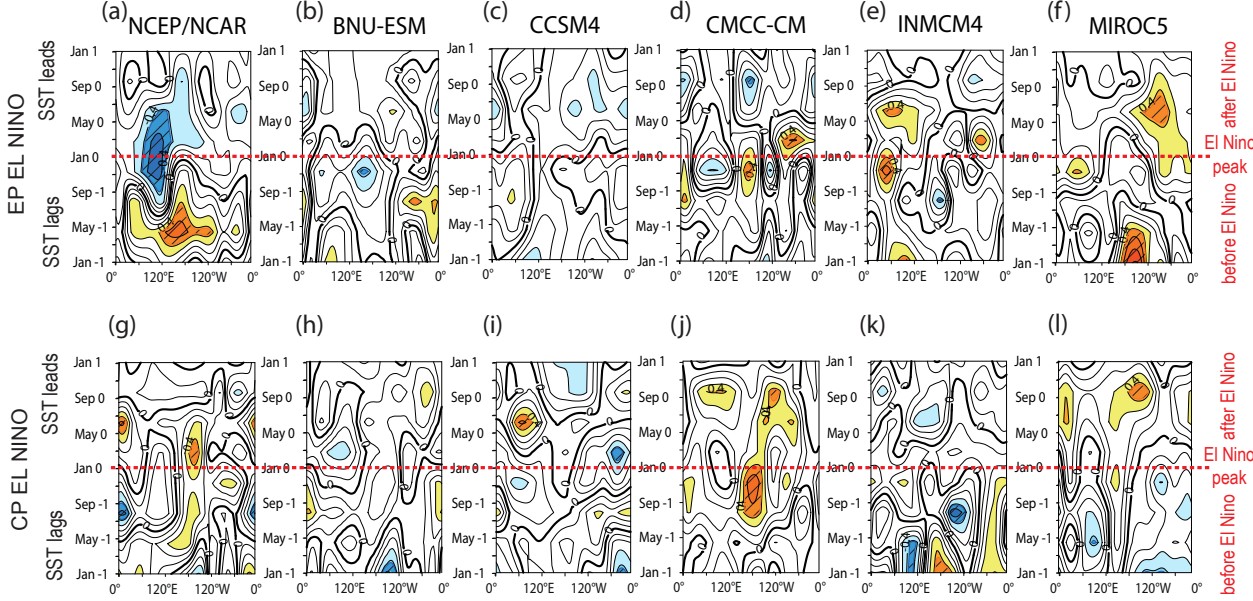

**Figure 9:** Time-longitude monthly lagged correlation of equatorial averaged (5°N–5°S) rms of MJO
filtered U850 and January E (a-f) and C (g-l) indices for NCEP/NCAR Reanalysis and 5 CMIP5 models.
Contour interval is 0.1. Negative correlation ≤–0.3 is blue shaded, positive correlation ≥0.3 is orange shaded.
Hatching lines denote correlation a 90% statistical confidence level based on a Gaussian statistics (≥ 0.42 and ≤-0.42).
The thick black line indicates the zero correlation line.

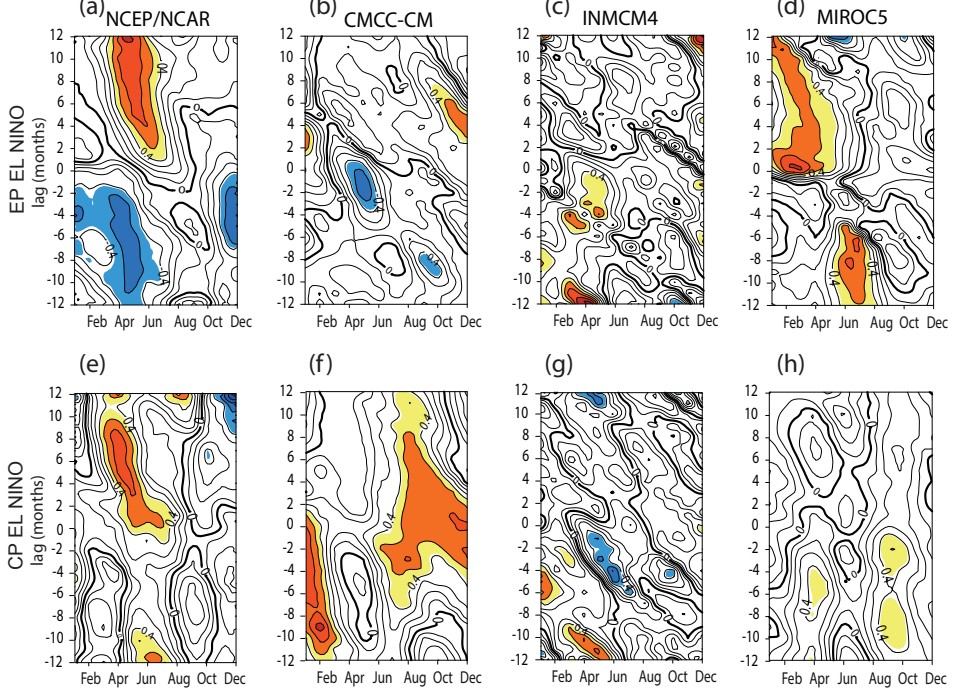

**Figure 10:** Monthly lagged correlation of E (a-d) and C (e-h) indices as a function of start month
with respect to MJO activity index WPacMJOu850 for NCEP/NCAR Reanalysis and 3 CMIP5 models.
Contour interval is 0.1. Negative correlation ≤–0.42 is blue shaded, positive correlation ≥0.42 is orange shaded.





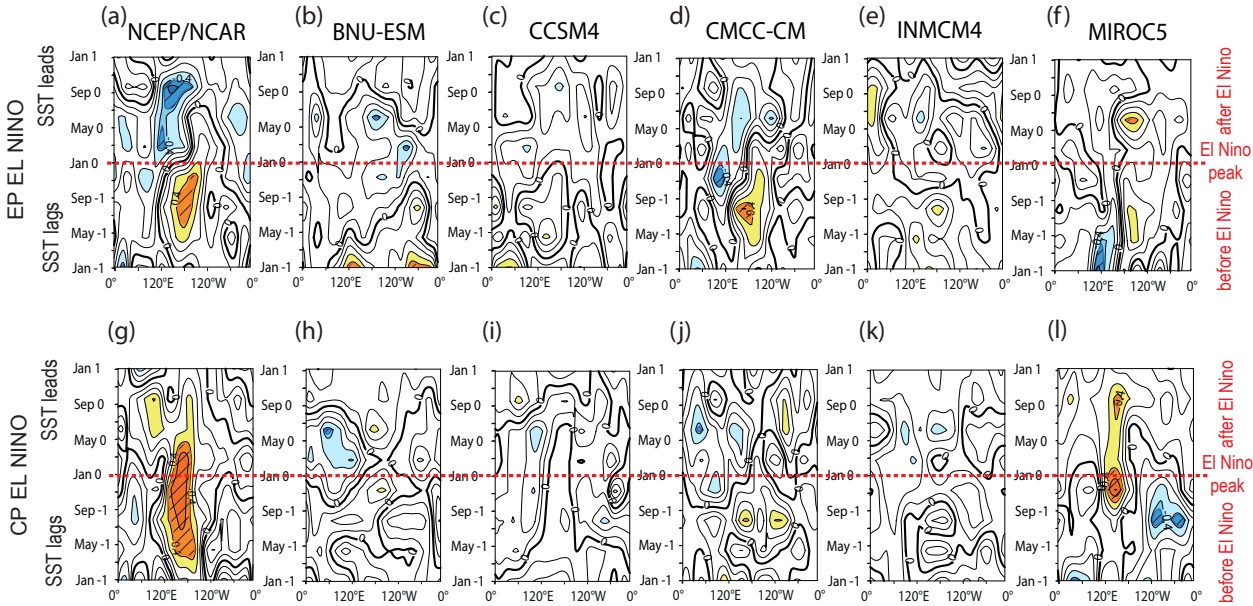

**Figure 11:** As Fig.9 but for Rossby waves

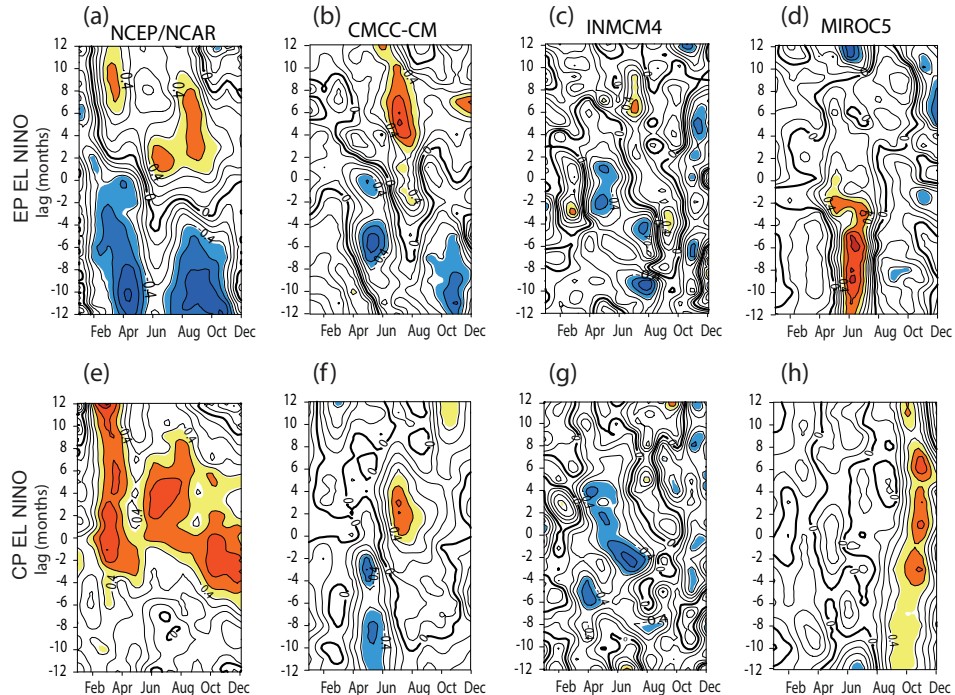

**Figure 12:** As Fig.10 but for Rossby waves activity index CPacERu850.