# Peer review of "The seasonal relationship between intraseasonal tropical variability and ENSO in CMIP5"

_Geoscientific Model Development, 2017_

## Referee Comment (RC1) · Anonymous Referee #1 · 8 Jun 2017

This is a straightforward manuscript: Signals of ENSO and tropical intraseasonal variability (ITV) in 23 CMIP5 models are compared and their possible relationship evaluated. The authors identified 16 models that reproduced two types of El Nino events, 5 that produced critical properties of ITV contributions to ENSO, and 4 that reproduced the MJO and equatorial Rossby waves. They concluded that only these latter 4 can be used to investigate the connection between ITV and ENSO in future climate.

These results are interesting and useful. The manuscript can be accepted for publication in GMD after some clarification and improvement. I recommend a major revision which I outline below.

1. There are some technical issues that need to be addressed. The band-pass filtering of zonal wavenumber k = 1 − 3 for the MJO and k = -1 − -8 for the Rossby wave are inappropriate for model simulations. According to Hayashi (1979), only the part of the eastward power that is incoherent with its equivalent westward power represents true eastward propagating signals. The coherence part represents stationary of standing signals. So using k = 1 − 3 to represent the MJO and k = -1 − -8 to represent the Rossby wave would exaggerate the propagating signals. In observations, the east-west equivalent signals are weak, so this practice is ok. For model simulations, such east-west equivalent signals are strong, the potential coherence part is great and this practice is problematic. The regression results from Jiang et al (2015, Fig. 3) clearly show the dominant stationary signals in many model simulations. The band-pass filtering method used in this current study would mistakenly extract propagating signals from these simulations when there is none.

2. Discussions of the results are mostly qualitative and subjective, heavily relying on visual impression. Suggest use quantitative measures to compare models and between models and observations.

3. Significance level of 90% is lower than commonly used 95% in modern literatures. Suggest use this high standard.

4. Using U850 to define the MJO and Rossby wave might be problematic. There are obviously other perturbations in the same frequency band of the Rossby wave (Fig. 3). Why not use precipitation as everyone else did? This would yield results that can be directly compared to others.

5. Some missing literature citations should be added:

Hendon et al. (2007) for seasonally varying relationship between MJO activity and the ENSO cycle Kessler et al. (1995) for MJO inducing the oceanic Kelvin wave in the Western Pacific Zhang and Gottschalck (2002) for MJO as a precursor of El Nino.

References:

Hayashi, Y. 1979: A generalized method of resolving transient disturbances into standing and traveling waves by space-time spectral analysis. J. Atmos. Sci., 36, 1017-1029.

Hendon, H.H., Wheeler, M.C. and Zhang, C., 2007. Seasonal dependence of the MJO–ENSO relationship. Journal of Climate, 20(3), pp.531-543.

Jiang, X., et al. (2015), Vertical structure and physical processes of the Madden-Julian oscillation: Exploring key model physics in climate simulations, J. Geophys. Res. Atmos., 120(10), 4718-4748, doi: Doi 10.1002/2014jd022375.

Kessler, W. S., M. J. McPhaden, K. M. Weickmann, 1995: Forcing of intraseasonal Kelvin waves in the equatorial Pacific. J. Geophys. Res., 100, 10 613–10 631.

Zhang, C., and J. Gottschalck, 2002: SST anomalies of ENSO and the Madden-Julian Oscillation in the equatorial Pacific. J. Climate, 15, 2429-2445.

---

## Author Comment (AC1) · 24 Jul 2017

We would like to thank the reviewer for his constructive comments. We have incorporated the appropriate changes into the revised manuscript and have made substantial revisions to improve the quality of the manuscript.

**Referee's comment:** There are some technical issues that need to be addressed. The band-pass filtering of zonal wavenumber k = 1 - 3 for the MJO and k = -1 - 8 for the Rossby wave are inappropriate for model simulations. According to Hayashi (1979), only the part of the eastward power that is incoherent with its equivalent westward power represents true eastward propagating signals. The coherence part represents stationary of standing signals. So using k = 1 - 3 to represent the MJO and k = -1 - -8 to represent the Rossby wave would exaggerate the propagating signals. In observations, the eastwest equivalent signals are weak, so this practice is ok. For model simulations, such east-west equivalent signals are strong, the potential coherence part is great and this practice is problematic. The regression results from Jiang et al (2015, Fig. 3) clearly show the dominant stationary signals in many model simulations. The band-pass filtering method used in this current study would mistakenly extract propagating signals from these simulations when there is none.

**Authors' response:** This study may be considered as an extension of previous investigation of ITV/ENSO relationship by (McPhaden et al., 2006, Hendon et al., 2007, Gushchina and Dewitte, 2011,2012) which based on the observation. We need to use the model's data to explore this kind of relationship in future climate. As a first step we need to evaluate the model's skill in simulation ENSO/ITV. In all aforementioned studies the method of (Wheeler and Kiladis, 1999) were used. To compare our results with those obtained from observation we preferred to use the same method of ITV filtering. Noteworthy this method was used for ITV analysis in the CMIP3 and CMIP5 models (Lin et al., 2006; Hung et al., 2013). However we agree with the reviewer that models tend to overestimate the coherent westward part of MJO. Therefore we have thoroughly examined the models chosen for further analysis (MIROC5, INMCM4, CMCC-CM) from this aspect.

 Firstly we have analyzed the spectrum of ITV in the models and compare them to Reanalysis (Fig.1). It is obviously seen that in MIROC5 and CMCC-CM the westward power (blue square) is much lower than eastward power (red square) in the MJO intervals (period 30-60 days, zonal wave number 1-3 (eastward) and -1...-3 (westward)). Moreover in Reanalysis the westward power is rather intensive as compare to westward power in the models. Exception is the INM-CM4 models where the westward power is equivalent to eastward one. To verify if the signals are coherent we made the further analysis.

Figure 1: Space-time spectrum of the 15°N-15°S symmetric component of U850 divided by the background spectrum from NCEP/NCAR Reanalysis and 3 CMIP5 models.

2. For MJO (Rossby) waves we recompose the westward (eastward) signal in the same frequency intervals but for the opposite sign of zonal wave numbers: -1...-3 for MJO and +1..+8 for Rossby waves (Fig.2-3). It may be seen that the amplitude of westward analogue of MJO is significantly lower as compare to eastward propagating patterns (except for INM-CM4). For Rossby waves the amplitude of eastward and westward propagating signal is comparable but the timing, spatial localization and speed of propagating signal differ significantly. To confirm quantitatively this suggestion we calculated the correlation between eastward and westward propagating signals (Table 1).

---

## Author Comment (AC2) · 24 Jul 2017

[revised manuscript text omitted]

10 Using U850 field to define the MJO and Rossby wave in current study is attributed to the mechanism involved in ITV/ENSO relationship. Namely the westerly wind anomalies associated to MJO and Rossby waves may trigger the downwelling intraseasonal Kelvin waves responsible to El Niño generation.

For correlation analysis between ITV activity and ENSO types the so-called E and C indices based on the linear combination of the first two EOFs of the SST anomalies in the tropical Pacific were constructed (Takahashi et al., 2011).

15 Whereas the E index accounts for the extreme El Niño events (EP type), the C index grasps the variability associated to the CP El Niño and La Niña events. These indices, independent by construction (i.e. their correlation is zero), can be successfully used for correlation analyses (see Takahashi et al. (2011) for more details).

**3 Results**

5

**3.1 The two flavors of El Nino**

20 As a first step, the models' skill in simulating ENSO diversity is assessed. Takahashi et al. (2011) demonstrated that the patterns of the first two EOF of SST anomalies in the tropical Pacific capture the structure of SST anomalies associated to the two types of El Niño, with the EOF1 maximum localized in the eastern Pacific (typical of EP event) and EOF2 maximum

observed in the central Pacific (typical of CP event). The two first EOF modes obtained from 23 CMIP5 models are presented on Fig. 1-2.

Almost all the models (except for CSIRO-Mk3) reproduce the SST anomalies pattern associated to the EP El Niño (Fig.1) with the maximum localized in the eastern equatorial Pacific. The main deficiency of the models is the westward shift of

- 5 anomaly maximum as compare to observation. Almost half of models are unable to simulate adequately the tripole structure of CP El Niño with maximum in the central Pacific bordered by two centers of opposite sign (Fig. 2). Noteworthy that most models tend to overestimate the variability associated to the first EOF as compare to observation. In opposite the EOF2 contribution to the total variability is lower in the model than in HadISST data (except for ACCESS1-3, CSIRO-Mk3, FIO-ESM и INMCM4). Therefore we may suggest that models tend to underestimate the CP El Niño contribution to the SST
- 10 variability in the tropical Pacific.

To demonstrate the models skill in reproducing ENSO period the spectrum of C and E El Niño indices is presented (Fig.1-2). Most models simulate several distinct spectral peaks in the interval 3-9 years. The E index peaks are shifted to the shorter periods (3-6 years) while CP El Niño is characterized by longer oscillation (5-8 years), which is in a good agreement with observation (4-5 and 5-6 years for EP and CP events respectively). The models tend to overestimate the spectral density

15 which is partially due to the longer investigated period for the model as compare to HadISST (250 and 135 years respectively).

Based on the EOF and spectral analysis the 16 models capable to reproduce the spatial structure and temporal variability of SST associated to the two distinct type of El Niño were chosen for further analysis: BNU-ESM, CanESM2, CCSM4, CESM1-CAM5, CMCC-CM, CNRM-CM5, EC-EARTH, FIO-ESM, GFDL-CM3, GFDL-ESM2M, GISS-E2-H, INMCM4,

20 IPSL-CM5A-MR, MIROC 5, MPI-ESM-LR, MRI-CGCM3. The models which represent the EOF maximum in off equatorial regions and the models that don't simulate the EOF2 maximum in the tropical Pacific were excluded from the analysis.

**3.2 Intraseasonal tropical variability**

The characteristics of the ITV are then documented with the focus on the key aspects for relationship between ENSO and ITV, namely the intensity, spatial structure, propagation characteristics, frequency and seasonal cycle of ITV component. The aforementioned characteristics primarily determine the efficiency of ITV contribution to the El Niño development.

- 5 The earlier studies have evidenced inaccurate simulation of MJO and CCEW in CMIP models (Guo et al., 2015; Jiang et al., 2015; Klingaman et al., 2015; Xavier et al., 2015), with CMIP5 demonstrated more skill (Hung et al., 2013) than CMIP3 generation (Lin et al., 2006). We tested the models reliable in reproducing two type of El Niño to simulate adequately the main ITV components: MJO and equatorial Rossby waves. We had to exclude from the analysis the models which do not present in open access the daily data of U850 required for ITV filtering. The 20-years period was used for the analysis as most of the models have not longer period of daily data. The following models were analyzed (with the period of model
- which of the models have not longer period of daily data. The following models were analyzed (with the period of model years indicated in the parenthesis): BNU-ESM (1979-1998), CanESM2 (2441-2460), CCSM4 (1089-1108), CMCC-CM (1850-1869), CNRM-CM5 (2385-2404), GFDL-CM3 (0001-0020), GFDL-ESM2M (0001-0020), INMCM4 (2090-2109), IPSL-CM5A-MR (1800-1819), MIROC 5 (2000-2019), MPI-ESM-LR (2016-2035), MRI-CGCM3 (2086-2105). The model outputs are compared to the NCEP/NCAR Reanalysis data for the period 1980-1999.
- To isolate the ITV component the raw wavenumber-frequency spectra were plotted following the method of Wheeler and Killadis (see Section 2). Figure 3 presents the space-time spectra normalized above the background spectra for symmetric component of U850 wind from the Reanalysis over 1980-1999 (Fig. 3 upper panel) and as simulated by CMIP5 models over a 20-year period. The resulting contours can be thought of as levels of significance, with peaks in the individual spectra that are 20% above the background shown as shaded. Superimposed upon these plots are the dispersion curves for odd meridional mode number of equatorial waves for various equivalent depths (h=8, 12, 25, 50, 90 and 200 m). The MJO appears as a prominent signal, especially in the symmetric spectra (the spectra for antisymmetric component is not shown). Most of the models simulate the MJO signal, but 8 among 12 overestimate the westward propagating signal against eastward one. Signals of the Kelvin waves are obviously identified in the symmetric spectra for observation. The main model deficiency in representation of Kelvin wave is the unrealistic spectral maximum around eastward wave numbers 6-8. The
- 25 simulated signal appears in both the symmetric and antisymmetric spectra and was shown to be associated to the mid latitude

[revised manuscript text omitted]

The propagation velocity and spatial patterns of MJO are adequately simulated by the models (Fig. 8, Table 3), with however faster propagation in CCSM4 (Fig. 8c) and shorter wave length and lower speed in BNU-ESM (Fig. 8b). The equatorial Rossby waves have larger longitudional extent in the models (Fig. 9). The propagation velocity is comparable to Reanalysis in BNU-ESM (Fig. 9b), CMCC-CM (Fig. 9d) and INM-CM4 (Fig. 9e), and is too low in CCSM4 (Fig. 9c) and MIROC5 (Fig. 9f). In INMCM4 and MIROC 5 the intensity of Rossby waves is overestimated.

Following Hayashi (1979) only the part of the eastward power that is incoherent with its equivalent westward power 20 represents true eastward propagating signals. Moreover the regression results from Jiang et al (2015) emphasize the dominant stationary signals in many model simulations. To verify if the stationary signal is presented in the models that were demonstrated to simulate realistically the variability and seasonal cycle of ITV components, we recomposed the signal in the same frequency intervals that for MJO and Rossby waves but for the opposite sign of zonal wave numbers: -1...-3 for MJO and +1..+8 for Rossby waves. Insignificant correlation between westward and eastward signals confirms that westward and

25 eastward parts are incoherent.

[revised manuscript text omitted]

- The discrepancies between observed and simulated ITV/ENSO relationships may be attributed to the fact that NCEP/NCAR Reanalysis data is compared to the PI-Control experiment. Recently Gushchina and Dewitte (2017) demonstrated the ITV/ENSO relationship is rather sensitive to the background state of the tropical Pacific system. Therefore over the investigated period (1979-2015) the mean state changes associated to the warming of tropical Pacific may influence the characteristic of observed ITV/ENSO relations, while the model data obtained from PI-Control experiment with preindustrial

concentration of greenhouse gases does not contain the mean state changes associated to the radiative forcing. The comparison of model data from Historical experiment, with mean state closer to observation, demonstrates better agreement with NCEP/NCAR Reanalysis (not shown). However the Historical experiment data does not continue beyond 2005, that limits the analysis of ITV behaviour during CP El Niño as CP events appear mostly in 2000s.

5 In spite of several discrepancies between the model and observation we may conclude that El Niño precursor signal induced by MJO and ER is presented in CMCC-CM, MIROC5 and INMCM4. This may involve further investigation of ITV predictive value relative to ENSO in future climate and its sensitivity to global warming using these models.

**Code availability**

The code on Fortran and Matlab is available on request from author (Tatiana Matveeva, matania.777@gmail.com).

**10 Acknowledgement**

This study is supported by grants of Russian Foundation of Basic Research No.15-05-06693 and No.16-35-00394\16. The study is carried out in frame of scientific program of Faculty of Geography of Moscow State University No.AAAA-A16-116032810086-4.

[revised manuscript text omitted]

---

## Referee Comment (RC2) · Anonymous Referee #2 · 26 Jul 2017

Twenty-three CMIP5 models are investigated for their match with observations in representing aspects of tropical intraseasonal and interannual variability. Despite the title, which emphasises the relationship between interannual and intraseasonal variability, the majority of the paper is first spent analysing which models are best at simulating individual aspects of the variability, namely the two types of ENSO, the MJO, and Equatorial Rossby and Kelvin waves. The results show a large variety of behaviour from the models, with very few models showing variability and relationships like observed. This may be of interest to model developers, but I don't think it adds much new insight into the dynamics of the observed variability. Also, I can't see how these results can help pin-point what aspects of the models need to be changed for improvement.

[Figure]

I understand this is a difficult task, but is one that needs to be done to help improve the models. At the very least I think this paper needs major revision. Some specific comments are as follows (listed in approximate order of appearance, not importance).

1. The English grammar needs improving to make it easier to read and understand. For example, there are many instances where the word "the" is inserted incorrectly or missing.

2. Page 3, line 24: Kim and You (2012) missing from reference list.

3. Page 5, line 6: "PI" is not defined.

4. Section 2.2: It is noteworthy that you are using zonal wind data instead of a proxy for clouds and convective rainfall (e.g. outgoing longwave radiation) as used by Wheeler and Kiladis (1999). This means that the variability highlighted by your wavenumber-frequency analysis (Figure 3) is somewhat different to that highlighted in Wheeler and Kiladis (1999). It also means that the variability you show and isolate is not necessarily 'convectively-coupled'. For example, Figure 3 indicates the existence of the global Rossby-Haurwitz waves for low westward-propagating wavenumbers and periods around 5 days. It also means that the convectively-coupled equatorial Rossby (ER) and Kelvin waves are much less clear in Figure 3. This means that your filtered fields will also contain a much greater mix of variability compared to Wheeler and Kiladis. Finally, I note that you use rectangles to define your regions of filtering instead of following the dispersion curves for the equatorial waves. Ideally you should change your fields and filtering to better match the characteristics of the waves. However, I support the use of the western Pacific wind indices later in the paper as this is consistent with the findings of Hendon et al. (2007).

5. How are the values in Table 3 calculated?

6. Page 13, lines 13-15. This is poor style for scientific writing. Please refer to this paper: http://onlinelibrary.wiley.com/doi/10.1029/2010EO450004/full

[Figure]

Robock, A. (2010), Parentheses Are (Are Not) for References and Clarification (Saving Space), Eos Trans. AGU, 91(45), 419–419, doi:10.1029/2010EO450004.

---

## Referee Comment (RC3) · Anonymous Referee #3 · 8 Aug 2017

General Comments

The authors attempt to relate the ability of CMIP5 coupled models to simulate ENSO with their ability to correctly simulate the seasonal cycle and coupling between atmospheric intraseasonal equatorial waves, namely the MJO and Equatorial Rossby (ER) waves and the ocean. While the observational support for such a relationship in the real world has been well-established by the authors and others, unfortunately most of models studied here appear to only marginally simulate such relationships. The physical relationship between the zonal wind variability and ENSO in the models has not been explored in detail, therefore in my opinion the paper should be revised to include

more diagnostics.

Specific Comments

The paper starts out with an interesting and useful analysis of the behavior of ENSO in the models. However, I did not get a sense from this manuscript of which aspects of the models lead to bad (or better) representation of ENSO. For example, no indication of the oceanic response to wind forcing has been shown. Is it possible that the ocean models might also be an issue? In the end, it seems to me that a paper like this one in a journal such as GMD should lead to some recommendations on how models could be improved. Have the authors checked any of the ocean data (TAO buoys, SODA reanalysis) as in Guschina and DeWitte (2011) to see whether the wind signals they are isolating are actually related to oceanic Kelvin waves? Otherwise trying to relate ITV to ENSO seems speculative. In a better model such as CMCC-CM there should be a realistic relationship between the wind forcing and the ocean response. I encourage the authors to expand on these points in a revision.

The other major issue has to do with the isolation of CCEWs. The authors are using broadly defined filters that are based on OLR or brightness temperature signals from satellite data. Based on the spectra in Fig. 3, there is little basis for using the filter bands they have chosen, which ultimately derive from precipitation signals. Talking about "waves" such as the MJO and ERs in Figs. 7 and 8 is very suspect, since filtering of just red noise will give you similar results. I suggest more diagnostics to establish the existence of zonal wind signals associated with the MJO and ER waves (see below).

Technical Comments:

Pg. 2, line 21: see also Keen, 1982 Mon. Wea. Rev. pg. 1405.

Pg. 3, line 14: "loose" => "lost"

Pg. 5, line 16: except that Lin et al. and Hung et al. used precipitation not wind, this

needs to be pointed out here.

Pg. 6, line 3: "the maximum of ITV/ENSO relationship is observed. ' It is not clear what you mean by this. Precisely how are the indices in Table 2 defined? Please provide more detail on this, perhaps by using the "NCEP-NCAR" data as an example, which should be the closest to reality. Also, only 4 models are shown in Table 2 yet other models are analyzed later.

Pg. 8, Line 8: The spectra in Fig. 3 should be replotted, since it is difficult to see the signals through the dispersion curves. In particular, the MJO peak should show be a wavenumber 1 signal but these are obscured by the dispersion lines.

A comparison with Hung et al. for those models that have overlap would be welcome. If I look for example at CanESM2 and CCSM4 spectra of rainfall in Hung et al., it seems that these two models have a good spectral peak for the Kelvin wave in rainfall, but there is no evidence for a corresponding zonal wind peak in Fig. 3. This just illustrates the problem with using wind to define the equatorial wave modes as used here.

Line 15: "lower" suggest "weaker"

Pg. 9, line 9: "The maximum. . ." I have no idea what this sentence means to say.

Pg. 10, line 1: I guess the periods chosen for Fig. 7 are chosen from random, but perhaps the authors looked for good examples from the reanalysis and each model? More detail is needed, including making the obvious but necessary point that the model fields in no way are expected to match the reanalysis or each other.

It is not clear where the statements on propagation velocity and intensity come from. These are only one year periods, and it seems that the authors are just making statements by visual comparisons between the plots. The characteristics of the waves in each model could be compared with reanalysis by using diagnostics of the type used by Wheeler et al. 2000 (J. Atmos. Sci. pg. 613) or Hung et al. (their Fig. 9). The characteristics of the "waves" identified here overwhelmingly determined by the filtering, which sets the phase speed in particular. The plots in Fig. 8 are great examples of getting "something from nothing" by filtering: The CCSM4 zonal wind spectrum in Fig. 3 shows no signal at all for ER waves, yet Fig. 8c shows lots of westward propagation, which must come primarily from the constraints of the filter. There are lots of other examples of this.

Pg. 11, top: Much more discussion of what the expected relationship between the zonal wind and ENSO is needed here. Although the authors refer back to Takahashi (2009) and their own previous work, it will not be immediately obvious what Figs. 9a and 10a imply for ENSO forcing. The caption for Fig. 9 does not help. A brief review of the concepts is needed at the start of Section 3.3.1 before Figs. 9 and 10 can be interpreted.

Unfortunately, the model results from Figs. 9 and 10 are not very impressive, with little statistical significance indicated. Also, it is difficult to even tell the sign of many of the signals, so I suggest using more color.

Pg. 11, line 9: Indian Ocean wind stress could not force ENSO.

Line 10: 9g => 9h

Line 21: Here it is difficult to even tell what the sign is in Fig. 10h.

Pg. 12, line 2: Puy et al. 2016

Line 15: I wouldn't say it's "very close", but certainly it's better than the rest. To be honest, since Figs. 10-12 are based on the models' own renditions of ENSO, the huge disparity between them leads to an obvious conclusion: what forces ENSO in most of these models is something different than what forces it in the real world. I think this is the statement you should make more forcefully.

Pg. 14, line 7: "The deficiency of INM-CM4…" Little basis for this statement is shown. More detailed diagnostics of the ocean response would bolster claims like this, even by using SST if sea surface height or thermocline depth cannot be obtained.

---

## Author Comment (AC3) · 23 Dec 2017

**Response to Reviewer #1.**

We would like to thank the reviewer for his constructive comments.

There are some technical issues that need to be addressed. The band-pass filtering of zonal wavenumber k = 1 - 3 for the MJO and k = -1 - 8 for the Rossby wave are inappropriate for model simulations. According to Hayashi (1979), only the part of the eastward power that is incoherent with its equivalent westward power represents true eastward propagating signals. The coherence part represents stationary of standing signals. So using k = 1 - 3 to represent the MJO and k = -1 - 8 to represent the Rossby wave would exaggerate the propagating signals. In observations, the eastwest equivalent signals are weak, so this practice is ok. For model simulations, such east-west equivalent signals are strong, the potential coherence part is great and this practice is problematic. The regression results from Jiang et al (2015, Fig. 3) clearly show the dominant stationary signals in many model simulations. The band-pass filtering method used in this current study would mistakenly extract propagating signals from these simulations when there is none.

In order to address the reviewer's comment, we have carried out additional analyses in order to check the importance of the east-west equivalent signals.

- The analysis of ITV spectrum (new figure 2) shows the strong westward signal in 5 models among 16 (CanESM2, CNRM-CM5, IPSL-CM5A-MR, MPI-ESM-LR, MRI-CGCM3). These models are excluded from further analysis. In other models the westward power is of the same order than in Reanalysis. Exception is the INM-CM4 models where the westward power is equivalent to eastward one. To verify if the signals are coherent we made the further analysis. Below we present the results for INM-CM4 and two other models for comparison.
- 2. We recompose the U850 signal in the same frequency intervals as for MJO and ER but for the opposite sign of zonal wave numbers: -1...-3 for MJO (westward propagation) and +1..+8 for Rossby waves (eastward propagation) (Figures A1 andA2). It may be seen that the amplitude of westward analogue of MJO is significantly lower as compare to eastward propagating patterns (except for INM-CM4). For Rossby waves the amplitude of eastward and westward propagating signal is comparable but the timing, spatial localization and speed of propagating signal differ significantly. To confirm quantitatively this suggestion we calculated the correlation between eastward and westward propagating signals (Table 1). The correlation is rather small that allows suggesting that the signals are incoherent.

---

## Author Comment (AC4) · 23 Dec 2017

**Response to Reviewer #2**

We appreciate the reviewer's constructive comments.
The following is our point-to-point reply to these comments.

Twenty-three CMIP5 models are investigated for their match with observations in representing aspects of tropical intraseasonal and interannual variability. Despite the title, which emphasises the relationship between interannual and intraseasonal variability, the majority of the paper is first spent analysing which models are best at simulating individual aspects of the variability, namely the two types of ENSO, the MJO, and Equatorial Rossby and Kelvin waves. The results show a large variety of behavior from the models, with very few models showing variability and relationships like observed. This may be of interest to model developers, but I don't think it adds much new insight into the dynamics of the observed variability. Also, I can't see how these results can help pin-point what aspects of the models need to be changed for improvement. I understand this is a difficult task, but is one that needs to be done to help improve the models.

We agree with the reviewer that our paper does not propose or suggest ways to improve the models. It is an evaluation of the realism of the ITV/ENSO relationship, and as such, the paper can be viewed as a preliminary step towards suggesting improvement in the model physics, considering that it proposes a physically-based metrics to evaluate models and classify them into two broad classes, the less and most realistic ones. Such a "classification" could be the basis for identifying differences in some key dynamical aspects of the ITV, like the energy sources of the ITV (i.e. extra-tropical disturbances, tropical instabilities or non-linear interactions of multiple waves), its coupling with SST, its seasonal phase locking etc.

Despite the limitation of not addressing issues on model development, we believe that our results may still fit with the scope of Geoscientific Model Development since it provides "*new methods for assessment of models, including work on developing new metrics for assessing model performance and novel ways of comparing model results with observational data*", as well as proposes "*novel ways of comparing model results with observational data*"

1. The English grammar needs improving to make it easier to read and understand.
For example, there are many instances where the word "the" is inserted incorrectly or missing.

We have thoroughly revisited the text and improved the English grammar

2. Page 3, line 24: Kim and You (2012) missing from reference list.

Added to the reference list

3. Page 5, line 6: "PI" is not defined.

Pre-Industrial – corrected

4. Section 2.2: It is noteworthy that you are using zonal wind data instead of a proxy for clouds and convective rainfall (e.g. outgoing longwave radiation) as used by Wheeler and Kiladis (1999). This means that the variability highlighted by your wavenumberfrequency analysis (Figure 3) is somewhat different to that highlighted in Wheeler and Kiladis (1999). It also means that the variability you show and isolate is not necessarily 'convectively-coupled'. For example, Figure 3 indicates the existence of the global Rossby-Haurwitz waves for low westward-propagating wavenumbers and periods around 5 days. It also means that the convectively-coupled equatorial Rossby (ER) and Kelvin waves are much less clear in Figure 3. This means that your filtered fields will also contain a much greater mix of variability compared to Wheeler and Kiladis. Finally, I note that you use rectangles to define your regions of filtering instead of following the dispersion curves for the equatorial waves. Ideally you should change your fields and filtering to better match the characteristics of the waves. However, I support the use of the western Pacific wind indices later in the paper as this is consistent with the findings of Hendon et al. (2007).

In the revised manuscript, we better justify the use of U850 field for deriving the ITV components:

"We use here the U850 field for deriving the various components of the ITV instead of Outgoing Longwave Radiation (OLR) or brightness temperature signals from satellite data noting that the regions in the frequency-wavenumber domains where the spectral energy peaks are similar for OLR and U850, which is also predicted by a simple dynamical model of ITV (Thual et al., 2014). Moreover the use of U850 eases the interpretation of the results since it is the westerly wind anomalies that serve a physical conduit from the ITV to the ENSO dynamics. This approach follows previous relevant studies (McPhaden et al. 2006; Hendon et al. 2007)."

In the paper, the focus is on two component of ITV – MJO and equatorial Rossby (ER) wave which were shown to be associated to El Niño development (McPhaden et al., 2006, Hendon et

al., 2007, Gushchina and Dewitte, 2011, 2012). The Figure B1 provides the wavenumber-frequency spectra for both OLR and U850 and it can be seen that, the domains where the MJO and ER spectral energy peaks are comparable for both fields. The MJO spectral maximum in OLR is shifted to the higher zonal wave numbers as compare to U850 in accordance with the results of previous studies (Zhang, 2005).

For example, Figure 3 indicates the existence of the global Rossby-Haurwitz waves for low westward-propagating wavenumbers and periods around 5 days. It also means that the convectively-coupled equatorial Rossby (ER) and Kelvin waves are much less clear in Figure 3. This means that your filtered fields will also contain a much greater mix of variability compared to Wheeler and Kiladis.

The frequency band for MJO filtering is 30-60 days and for ER – 10-50 days, thus we do not include in our filtered fields the waves with 5 days period. Overall the main difference between OLR and U850 spectra is located at the periods shorter than 10 days, which does not impact our results. We have improved the presentation of figure 3 (new figure 2) so as to better visualize the MJO and ER domains. We applied the same color scale as in Hung et al. (2013) for easing the comparison with their results.

Note that we use rectangles to define our regions of filtering only for MJO as in Wheeler in Kiladis (1999). For Rossby wave following (Wheeler and Kiladis, 1999) we follow the dispersion curves for the equatorial waves with equivalent depth ranging from 8 to 90 m. This is now mentioned in the text of the revised manuscript. "For Rossby waves, the frequency-wavenumber bands is also limited by the dispersion curves corresponding to values of the atmosphere equivalent depth ranging from 8 m to 90 m, which follows (Wheeler and Kiladis, 1999)"

[Figure]

**Figure B1: Space–time spectrum of the 15°N–15°S symmetric component of U850 (upper panel) and OLR (bottom panel) divided by the background spectrum for CMCC-CM model**

5. How are the values in Table 3 calculated?

We agree with the reviewer that the estimates presented in Table 3 are rather uncertain. We have removed this Table and added to the revised version a figure that is comparable to figure 9 of Hung et al. (2013) and that indicate the main values of expected phase speed (figures 7-8).

6. Page 13, lines 13-15. This is poor style for scientific writing. Please refer to this paper: http://onlinelibrary.wiley.com/doi/10.1029/2010EO450004/full

We have improved the style, which should ease the readability of the revised manuscript.

---

## Author Comment (AC5) · 23 Dec 2017

We thank the reviewer for his/her constructive comments, which has motivated us to improve our manuscript. Please, see our response in the attached (supplement) file.

Please also note the supplement to this comment: https://www.geosci-model-dev-discuss.net/gmd-2017-92/gmd-2017-92-AC5-supplement.pdf